# Amiselimod (MT-1303), a novel sphingosine 1-phosphate receptor-1 functional antagonist, inhibits progress of chronic colitis induced by transfer of CD4+CD45RBhigh T cells

**Kyoko Shimano, Yasuhiro Maeda, Hirotoshi Kataoka, Mikako Murase, Sachiko Mochizuki, Hiroyuki Utsumi, Koichi Oshita, Kunio Sugahara**⊙ *

Sohyaku, Innovative Research Division, Mitsubishi Tanabe Pharma Corporation, Yokohama, Japan

* Sugahara.Kunio@me.mt-pharma.co.jp

## Abstract

Amiselimod (MT-1303) is a novel sphingosine 1-phosphate receptor-1 (S1P$_1$ receptor) modulator with a more favorable cardiac safety profile than other S1P$_1$ receptor modulators. MT-1303 phosphate (MT-1303-P), an active metabolite of MT-1303, exhibits S1P$_1$ receptor agonism at a lower EC$_{50}$ value than other S1P$_1$ receptor modulators currently being developed. We aimed to evaluate the efficacy of MT-1303 and its mode of action in chronic colitis using an inflammatory bowel disease (IBD) model. Oral administration of MT-1303 (0.3 mg/kg) once daily for 3 days to mice almost completely abolished S1P$_1$ receptor expression on CD4$^+$ T cells from mesenteric lymph nodes, which corresponded to a marked decrease in CD4$^+$ T cell count in peripheral blood, indicating that MT-1303-P acts as a functional antagonist of the S1P$_1$ receptor. The potential benefit of MT-1303 for IBD was assessed using immunodeficient SCID mice with chronic colitis induced by adoptive transfer of CD4$^+$CD45RB$^{high}$ T cells from BALB/c mice. An oral dose of 0.1 and 0.3 mg/kg MT-1303 administered daily one week after the cell transfer inhibited the development of chronic colitis with an efficacy comparable to that of an anti-mTNF-α mAb (250 μg/mouse). In addition, MT-1303 administration significantly reduced the number of infiltrating Th1 and Th17 cells into the lamina propria of the colon in colitis mice. Our results suggest that MT-1303 acts as a functional antagonist of the S1P$_1$ receptor on lymphocytes, regulates lymphocyte trafficking, and inhibits infiltration of colitogenic Th1 and Th17 cells into the colon to inhibit the development of chronic colitis.

## Introduction

Inflammatory bowel disease is a general term for diseases involving chronic persistent colitis, and is represented by ulcerative colitis (UC) and Crohn's disease (CD). While the etiology of inflammatory bowel disease remains unknown, evidence suggests that it may be caused by dysregulation of mucosal immune responses to environmental factors such as food antigens and

**Data Availability Statement:** All relevant data are within the manuscript and its Supporting Information files.

**Funding:** Funding: Mitsubishi Tanabe Pharma Corporation funded these studies. The funder provided support in the form of salaries and research materials for authors, Kyoko Shimano, Yasuhiro Maeda, Hirotoshi Kataoka, Mikako Murase, Sachiko Mochizuki, Hiroyuki Utsumi, Koichi Oshita, and Kunio Sugahara, but did not have any additional role in the study design, data collection and analysis, decision to publish, or preparation of the manuscript. The specific roles of these authors are articulated in the 'author contributions' section.

**Competing interests:** Competing interests: KS, YM, HK, MM, SM, HU, KO, and KS are employees of Mitsubishi Tanabe Pharma Corporation. This does not alter the authors' adherence to PLOS ONE policies on sharing data and materials. MT-1303 is the product in development, and Mitsubishi Tanabe Pharma Corporation has the ownership of the patents of MT-1303.

enteric bacterial flora [1]. One of the histopathological features of UC and CD is markedly increased numbers of leukocytes, specifically memory T cells, in affected tissues [2,3]. Such dysregulated leukocyte recruitment is likely a result of increased cell extravasation and/or retention in these sites. Recent basic and clinical research indicate that the targeting of molecules involved in leukocyte trafficking may be a promising therapeutic strategy in IBD [4].

Amiselimod (MT-1303) is an oral selective sphingosine 1-phosphate receptor-1 ($S1P_1$ receptor) modulator [5] that is converted to its active metabolite, MT-1303 phosphate (MT-1303-P), by sphingosine kinases in vivo. A phase I study demonstrated that MT-1303 has a more favorable cardiac safety profile than fingolimod, the first $S1P_1$ receptor modulator approved for the treatment of relapsing-remitting multiple sclerosis (RRMS) [6]. A phase II study that enrolled more than 400 patients with RRMS reported that MT-1303 at doses up to 0.4 mg had superior efficacy over the placebo control as well as a benign safety profile [7].

$S1P_1$ receptor modulators induce $S1P_1$ receptor internalization in lymphocytes, inhibit lymphocyte egress from secondary lymphoid organs by reducing the S1P responsiveness of lymphocytes, and consequently exert immunomodulatory effects by markedly reducing the infiltration of T cells into inflamed sites [8]. Given the potential role of the $S1P_1$ receptor in the pathogenesis of immune-mediated diseases, the S1P-$S1P_1$ receptor axis may be a potential target for the treatment of IBD. In fact, studies have reported that $S1P_1$ receptor modulators effectively ameliorate the severity of symptoms in dextran sulfate sodium (DSS)-induced colitis, 2,4,6-trinitrobenzenesulphonic acid (TNBS)-induced colitis, adoptive $CD4^+CD45RB^{high}$ T cell transfer colitis and interleukin (IL)-10 knockout mice, murine models of experimental colitis [9–11]. Moreover, in a phase II trial in patients with moderate-to-severe UC, a daily dose of 1 mg ozanimod resulted in a higher rate of clinical remission at 8 weeks, the primary endpoint, than placebo [12].

In the present study, we evaluated the effects of MT-1303 on chronic colitis in immunodeficient SCID mice induced by adoptive transfer of $CD4^+CD45RB^{high}$ T cells from BALB/c mice, an animal model of IBD. These effects were compared with those of an anti-mouse TNF-α monoclonal antibody, and the mode of action of MT-1303 was examined in this animal model. In addition, we assessed and compared the $S1P_1$ receptor agonist activity of the active metabolite MT-1303-P with that of other $S1P_1$ receptor modulators that are now under development, and confirmed that MT-1303 acts as a functional antagonist of lymphocytic $S1P_1$ receptors in mice.

## Materials and methods

### Mice

Female BALB/c mice (n = 252) and male C57BL/6 mice (n = 6) aged 5 weeks were purchased from Charles River Laboratories Japan (Yokohama, Japan) and female SCID mice (n = 174) aged 5 weeks were purchased from CLEA Japan (Tokyo, Japan). Mice were housed 3–5 per stainless cage under specific pathogen-free conditions. They were kept at a constant temperature of 23±3°C and relative humidity of 30–70% under a 12-h light/dark cycle. Food and water were available *ad libitum*. All animal experiments were performed using experimental protocols approved by the ethics review committee for animal experimentation of the Research Division of Mitsubishi Tanabe Pharma (Application Number: BJ10-0073/74, BJ12-0822/0823/0824/0826, BJ14-0678/0679). All mice were euthanized by isoflurane anesthesia followed by exsanguination. All efforts were made to minimize animal suffering, and animal health was carefully monitored every day. There were no animals euthanized at the defined humane endpoint (development of clinical disease, ruffled fur, lethargy, decreased body temperature).

## Agents and antibodies

Amiselimod (MT-1303; 2-amino-2-{2-[4-(heptyloxy)-3-(trifluoromethyl)phenyl] ethyl}pro-pan-1,3-diol hydrochloride), (S)-MT-1303 phosphate (MT-1303-P), siponimod [13], ozani-mod [11] and etrasimod [14] were provided by Mitsubishi Tanabe Pharma Corporation (Osaka, Japan). S1P was obtained from Avanti Polar Lipids (Alabaster, AL, USA). Anti-mouse TNF-α (mTNF-α) monoclonal antibody (mAb) (TN3-19.12), R-phycoerythrin (PE)-conju-gated anti-mouse CD45RB mAb (16A), fluorescein isothiocyanate (FITC)-conjugated anti-mouse IFN-γ mAb (XMG1.2), PE-conjugated anti-mouse IL-17 mAb (TC11-18 H10.1) and FITC or Cy-Chrome-conjugated anti-mouse CD4 mAb (GK1.5) were purchased from BD Bio-sciences. Rat anti-mouse $S1P_1$ antibody (713412) and mouse $CD4^+$ T cell enrichment column were obtained from R&D Systems, and the Lamina Propria Dissociation Kit was purchased from Miltenyi Biotec.

## Cyclic AMP assay

Human $S1P_1$-expressing CHO cells (catalog number: 93-0207C2) were purchased from Euro-fins DiscoverX (Fremont, CA, USA)[15], and were cultured in Ham's F-12 nutrient mix medium (Thermo Fisher Scientific) supplemented with 10% fetal bovine serum (Thermo Fisher Scientific), 100 U/mL penicillin, 100 μg/mL streptomycin, 300 μg/mL hygromycin B and 800 μg/mL geneticin. The cAMP Gi Kit (Cisbio) was used for the cAMP assay. The test substance was dissolved in 80% ethanol containing 10 mM NaOH and diluted in Hanks' bal-anced salt solution (HBSS; Life Technologies) containing 0.1% fatty acid-free bovine serum albumin (BSA; Sigma-Aldrich). The cells were seeded in 96-well half-area plates ($1.65 \times 10^4$ cells in 25 μL, per well) with 1 mM 3-isobutyl-1-methylxanthine. A 20-μL volume of test sub-stance solution and 5 μL of 10 μM forskolin solution were added, and the cells were incubated for 45 minutes at room temperature. Subsequently, homogenous time-resolved fluorescence reagents (Cisbio) were added to the plate and incubated for 1 hour at room temperature. The ratio of the acceptor and donor emission signals (ratio = signal 665 nm/signal 615 nm x $10^4$) was measured in each individual well using an EnVision plate reader (PerkinElmer). The aver-age was taken from three independent experiments with each experiment run in duplicate and expressed as a percentage of the maximum response to S1P.

## Measurement of $S1P_1$ expression on $CD4^+$ T cells from lymph nodes

After oral administration of MT-1303 at 0.3 mg/kg once daily for 3 days, blood and mesenteric lymph nodes were collected from C57BL/6 mice under isoflurane anesthesia. Single-cell sus-pensions were prepared from the mesenteric lymph nodes. Expression of the $S1P_1$ receptor on lymphocytes was examined by incubating the cell suspensions with rat anti-mouse $S1P_1$ mAb or rat IgG2a isotype control (eBR2a; eBioscience) at 40 μg/mL in flow cytometry staining buffer solution (FCS buffer; eBioscience) containing 1 mM EDTA and 2% normal mouse serum on ice in the dark for 1 h. The cells were then washed with PBS and incubated with bio-tin-conjugated anti-rat IgG antibody (Jackson ImmunoResearch Laboratories) at 7 μg/mL for 20 min, followed by PE-conjugated streptavidin (eBioscience) at 1 μg/mL for 20 min. Cell sus-pensions were subsequently incubated with 2% normal rat serum in FCS buffer for 20 min, fol-lowed by FITC-conjugated anti-mouse CD4 mAb for 20 min. After washing, the cells were resuspended in FCS buffer for flow cytometric analyses (LSR, BD Biosciences). Collected peripheral blood (0.1 mL) was hemolyzed and fixed using the IMMUNOPREP™ Reagent Sys-tem (Beckman Coulter). The number of lymphocytes, T cells, and $CD4^+$ T cells was measured using a Cytomics™ FC500 flow cytometer (Beckman Coulter) with Flow-Count™ (Beckman Coulter) as the internal standard.

## Generating the colitis model and determining clinical scores

Mesenteric lymph nodes were harvested from donor BALB/c mice after euthanasia, and CD4$^+$ T cells were isolated using the CD4$^+$ column (R&D Systems). The CD4$^+$ T cells were labeled with PE-conjugated anti-mouse CD45RB mAb and sorted using a MoFlo cell sorter (Cytomation MoFlo Sorter; Beckman Coulter) to obtain the CD45RB$^{high}$ fraction. Each recipient SCID mouse was injected through the tail vein with $1.25 \times 10^5$ CD4$^+$CD45RB$^{high}$ T cells to induce colitis.

To evaluate the prophylactic effects of MT-1303, SCID mice were divided into 4 groups on the day of cell transfer using the simulation method so that the mean and variance of body weight was approximately equal among the groups. The groups were 1) vehicle-treated control, 2) MT-1303 0.1 mg/kg, 3) MT-1303 0.3 mg/kg (n = 15 each), and 4) no cell transfer normal group (n = 14). For comparison of the effects of MT-1303 and the anti-mTNF-α mAb, SCID mice that received the cell transfer were pooled and divided into 4 groups (n = 18 each) on day 7 after cell transfer using the simulation method so that the mean and variance of the body weight ratio and body weight were approximately equal among the groups 7 days after cell transfer. The groups were 1) vehicle, 2) MT-1303 0.1 mg/kg, 3) MT-1303 0.3 mg/kg, and 4) anti-mTNF-α mAb group. To evaluate the effect of MT-1303 on the infiltration of CD4$^+$ T cells into the colon, SCID mice that received the cell transfer were divided into 2 groups on day 7 after cell transfer: 1) vehicle (n = 11) and 2) MT-1303 0.3 mg/kg (n = 12). MT-1303 0.1 and 0.3 mg/kg groups received MT-1303 dissolved in 0.5% hydroxypropylmethyl cellulose (HPMC) solution orally every day, while the no cell transfer normal and vehicle-treated control groups were administered 0.5% HPMC solution alone. Infliximab, an anti-human TNF-α mAb, is administered at a dose of 5 mg/kg to patients with CD 2 and 6 weeks after the first treatment and at 8-week intervals thereafter. When an attenuation of the inhibitory effect of infliximab is observed in CD patients, the dose is increased to 10 mg/kg [16]. In this study, the anti-mTNF-α mAb was intraperitoneally injected at a dose of 250 μg/mouse, corresponding to approximately 10 mg/kg, on days 7 and 21 after cell transfer.

The body weight of SCID mice was monitored daily, and the ratio (%) of the body weight on each day to that on the day of cell transfer was calculated for each mouse. The day after the final administration (day 28 after T cell transfer), colitis was assessed using clinical scores determined by summing the scores for four parameters: hunching, wasting, colon thickening, and stool consistency. The score for each of the 4 parameters was determined as follows: hunching and wasting, 0 or 1; colon thickening, 0–3 (0, no colon thickening; 1, mild thickening; 2, moderate thickening; and 3, extensive thickening); and stool consistency, 0–3 (0, normal beaded stool; 1, soft stool; 2, diarrhea; 3, gross bloody stool) [17,18].

## Histology of colons from SCID mice

Paraffin sections of 3-μm thickness were prepared from the colons of SCID mice, fixed in 10% formalin solution and stained with hematoxylin and eosin.

## Intracellular cytokine staining and analysis

Colons were extracted from colitis-induced mice after euthanasia, and lamina propria lymphocytes (LPLs) were isolated using the Lamina Propria Dissociation Kit and gentleMACS$^{TM}$ Octo Dissociator according to the manufacturer's instructions (Miltenyi Biotec). For intracellular cytokine staining, the cells were cultured in the presence of brefeldin A solution, 50 ng/mL PMA and 1000 ng/mL ionomycin in RPMI 1640 medium containing 10% fetal calf serum, 10 mM HEPES, 100 U/mL penicillin, 100 μg/mL streptomycin and 50 μM 2-mercaptoethanol for 6 h at 37˚C in 5% CO$_2$. After the incubation, intracellular cytokine staining was performed

using anti-CD4, anti-IFN-$\gamma$ and anti-IL-17 mAbs. Flow cytometric analysis was conducted using LSR with CellQuest software (Becton Dickinson).

To measure cytokine production by LPLs, the cells were stimulated with PMA and ionomycin in RPMI 1640 medium containing 10% fetal calf serum, 10 mM HEPES, 100 U/mL penicillin, 100 $\mu$g/mL streptomycin and 50 $\mu$M 2-mercaptoethanol for 6 h at 37˚C in 5% $CO_2$. After the stimulation, levels of IFN-$\gamma$ and IL-17 secreted into the culture medium were measured using cell-based assays with a BD FACSArray Bioanalyzer (Becton Dickinson).

## Statistical analyses

The results are expressed as the mean ± S.E.M. Differences in clinical scores between groups were analyzed using Steel's test or the Wilcoxon test. Statistical differences in the body weight ratios (%) of body weight measured from the day of group allocation to the day after the final administration to body weight measured on the day of cell transfer were analyzed using Student's $t$-test or Dunnett's test. Differences in $S1P_1$ expression, the number of cells and cytokine production were analyzed using Student's $t$-test. Differences were considered significant at $p < 0.05$.

## Results

### $S1P_1$ agonist activity of MT-1303-P

The agonist activity of MT-1303-P on the human $S1P_1$ receptor was assessed by comparing its inhibition of forskolin-induced cAMP production with that of other $S1P_1$ receptor modulators, namely siponimod, ozanimod and etrasimod. MT-1303-P, siponimod, ozanimod and etrasimod showed $S1P_1$ receptor- and concentration-dependent inhibition of cAMP production, with $EC_{50}$ values of 0.013, 0.078, 0.33, and 0.57 nM, respectively (Table 1). MT-1303-P was the most potent of these compounds, with an $EC_{50}$ value approximately 5 times lower than that of siponimod and more than 20 times lower than that of ozanimod and etrasimod.

### MT-1303 administration induces $S1P_1$ receptor internalization in lymphocytes

After oral administration of MT-1303 at 0.3 mg/kg once daily for 3 days, flow cytometry was used to determine the $S1P_1$ expression on CD4[+] T cells from mesenteric lymph nodes of C57BL/6 mice and the number of CD4[+] T cells in peripheral blood. The day after the final dose, $S1P_1$ expression on CD4[+] T cells from mesenteric lymph nodes was significantly lower in the MT-1303-treated group than the control group (Fig 1A and 1B). Correspondingly, the

**Table 1. Agonist activity of S1P, MT-1303-P, siponimod, ozanimod and etrasimod on the human $S1P_1$ receptor.**

| Compound | $EC_{50}$[a] (95% CI), nM | $E_{max}$[b], mean ± S.E.M. % |
|---|---|---|
| S1P | 0.29 (0.17–0.48) | 100.0 ± 1.0 |
| MT-1303-P | 0.013 (0.0025–0.071) | 100.6 ± 0.4 |
| Siponimod | 0.078 (0.044–0.14) | 98.9 ± 0.6 |
| Ozanimod | 0.33 (0.14–0.82) | 99.0 ± 0.3 |
| Etrasimod | 0.57 (0.30–1.1) | 99.8 ± 0.6 |

[a]The $E_{max}$ values for each test substance were calculated at a fixed $E_{max}$ value for S1P of 100%.

[b]The $EC_{50}$ values for each test substance were calculated using nonlinear regression analysis.

Results were obtained from three independent experiments [S1P (n = 6), MT-1303-P, siponimod, ozanimod and etrasimod (n = 3)].

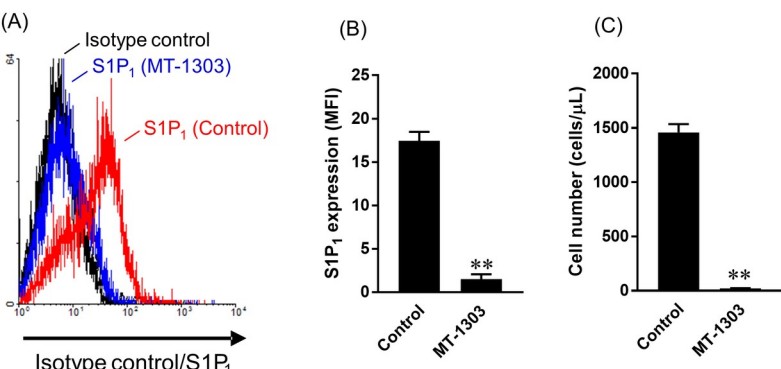

**Fig 1. Effects of MT-1303 on $S1P_1$ expression on $CD4^+$ T cells from mesenteric lymph nodes in mice.** MT-1303 (0.3 mg/kg) or vehicle was orally administered to C57BL/6 mice for 3 days and $S1P_1$ receptor expression on $CD4^+$ T cells from mesenteric lymph nodes was determined using flow cytometry. (A) Representative histograms of $S1P_1$ receptor expression in each group. Black, isotype control (rat IgG); red, vehicle-treated mice ($S1P_1$); blue, MT-1303-treated mice ($S1P_1$). (B) Mean fluorescence intensity (MFI) of the $S1P_1$ receptor on $CD4^+$ T cells. Data are expressed as the mean ± S.E.M. (n = 3), and statistical differences were calculated using Student's *t*-test after logarithmic transformation. **$p < 0.01$, compared with the vehicle-treated group. (C) The number of $CD4^+$ T cells in peripheral blood was measured using flow cytometry. Data are expressed as the mean ± S.E.M.; n = 3. Statistical differences were calculated using Student's *t*-test. **$p < 0.01$, compared with the vehicle-treated group.

number of $CD4^+$ T cells in peripheral blood was also markedly lower in the MT-1303-treated group than in the control group (Fig 1C).

## MT-1303 inhibits the development of colitis in SCID mice induced by adoptive transfer of $CD4^+CD45RB^{high}$ T cells

We evaluated the effect of MT-1303 on colitis induced by the adoptive transfer of $CD4^+CD45RB^{high}$ T cells from BALB/c mice to SCID mice. MT-1303 at doses of 0.1 and 0.3 mg/kg or vehicle was orally administered once daily for 28 days from the day of cell transfer. Vehicle-treated mice showed progressive weight loss from approximately 2 weeks after the cell transfer. Assessment of colitis using the sum of scores for hunching, wasting, colon thickening, and stool consistency showed that the clinical score (3.2 ± 0.6) in the control group was significantly higher than that in the normal group, indicating the development of colitis in the control group (Fig 2A and 2B). In contrast, body weight (%) in all MT-1303-treated groups remained higher than that in the control group from 4 days after cell transfer (Fig 2A). In particular, body weight (%) in the 0.3 mg/kg MT-1303-treated group was significantly higher than that in the control group from 17 days after cell transfer, and increased over time at a similar rate to the normal group. Additionally, clinical scores in the 0.1 and 0.3 mg/kg MT-1303-treated groups were 1.5 ± 0.4 and 1.1 ± 0.2, respectively, which were lower than those in the control group (Fig 2B). As shown in Fig 2C, histological analyses demonstrated the presence of inflammatory cell infiltrates, epithelial hyperplasia and mucin depletion from goblet cells in the colon of mice in the vehicle-treated group. In contrast, oral administration of MT-1303 at 0.3 mg/kg reduced inflammatory cell infiltrates, epithelial hyperplasia and mucin depletion from goblet cells in the colon.

## MT-1303 shows comparable efficacy to anti-mTNF-α mAb in colitis mice

Next, we evaluated the effect of MT-1303 and an anti-mTNF-α mAb on colitis induced by adoptive transfer of $CD4^+CD45RB^{high}$ T cells. MT-1303 (0.1 or 0.3 mg/kg) or vehicle was orally administered once daily for 21 days starting from 7 days after cell transfer, and the anti-mTNF-α mAb (250 μg/mouse) was intraperitoneally injected on days 7 and 21. Body weight

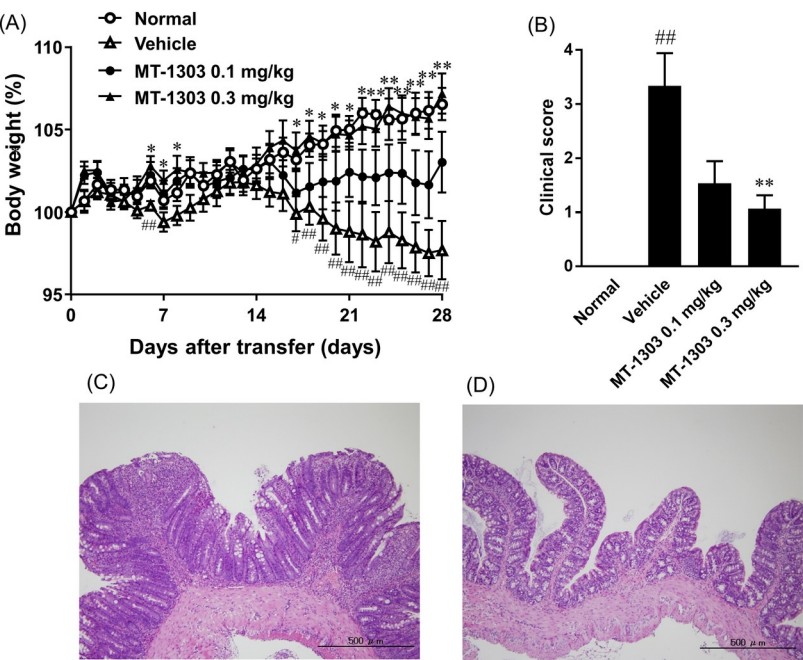

**Fig 2. Prophylactic effects of MT-1303 on colitis in SCID mice induced by adoptive transfer of CD4+CD45RB^high T cells.** SCID mice were injected with CD4+CD45RB^high T cells to induce colitis. MT-1303 or vehicle was orally administered to SCID mice every day from the day of CD4+CD45RB^high T cell transfer for 28 days. (A) Change in body weight over time. Body weight on each day is expressed as a percentage of the original weight. Each symbol represents the mean ± S.E.M. of body weight (%) in 14–15 mice (n = 14: normal group). Statistical differences were determined using Student's *t*-test by comparing to the normal group (#$p < 0.05$, ##$p < 0.01$) or using Dunnett's test by comparing with the vehicle-treated group (*$p < 0.05$, **$p < 0.01$). (B) Clinical scores were determined the day after the final administration. Each column represents the mean ± S.E.M. of clinical scores in 14–15 mice. Statistical differences were determined using the Wilcoxon test by comparing with the normal group (##$p < 0.01$) or using Steel's test by comparing with the vehicle-treated group (**$p < 0.01$). (C, D) Colon sections from vehicle- (C) or MT-1303 0.3 mg/kg (D)-treated mice were stained with hematoxylin-eosin.

(%) in the control group gradually decreased from 14 days after cell transfer (Fig 3A and 3B). Body weights (%) in the MT-1303- and anti-mTNF-α mAb-treated groups were higher than those in the control group from day 15. The clinical score in the control group was 3.6 ± 0.3, which was significantly higher than that in the normal group, confirming the development of colitis (Fig 3C). In contrast, clinical scores in the 0.1 and 0.3 mg/kg MT-1303-treated groups and anti-mTNF-α mAb-treated group were 1.1 ± 0.4, 0.6 ± 0.3, and 1.2 ± 0.3, respectively, which were significantly lower than those in the control group (Fig 3C). These results show that MT-1303 and the anti-mTNF-α mAb inhibit the development of colitis in SCID mice induced by adoptive transfer of CD4+CD45RB^high T cells from BALB/c mice, and that the inhibitory efficacy of MT-1303 is comparable to that of the anti-mTNF-α mAb.

## MT-1303 reduces infiltration of Th1 and Th17 cells into the colon of colitis mice

To elucidate the effect of MT-1303 on infiltration of CD4+ T cells into the colon of colitis mice, lymphocytes that had infiltrated the colons of colitis mice administered MT-1303 or vehicle for 3–4 weeks were analyzed using flow cytometry. The vehicle-treated group showed evidence of infiltration of lymphocytes including CD4+ T cells into the lamina propria of the colitic colon (Fig 4A and 4B). In contrast, treatment with MT-1303 significantly reduced the number of infiltrating lymphocytes and CD4+ T cells, with the CD4+ T cell count decreasing

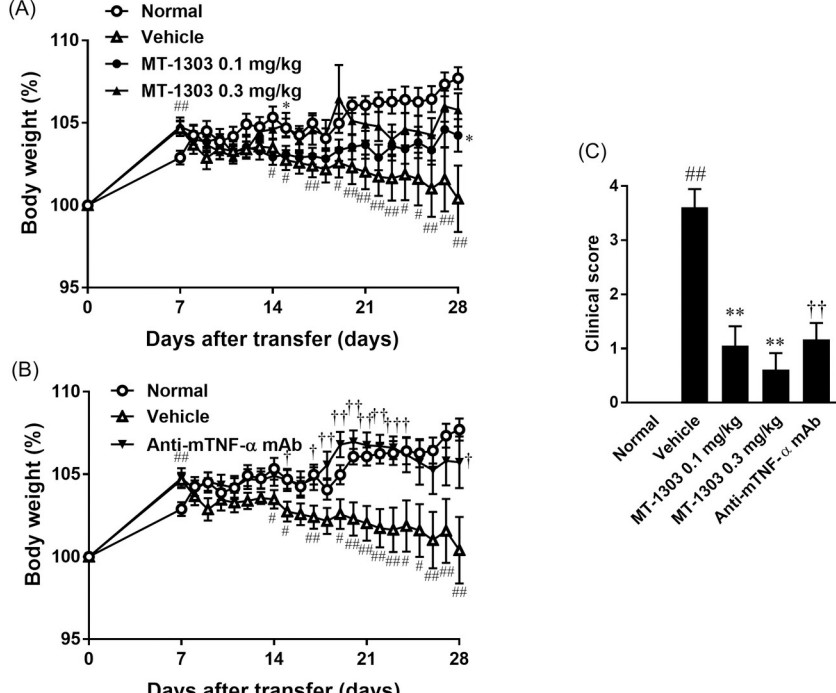

**Fig 3. Effects of MT-1303 and an anti-mTNF-α mAb on colitis in SCID mice induced by adoptive transfer of CD4$^+$CD45RB$^{high}$ T cells.** SCID mice were injected with CD4$^+$CD45RB$^{high}$ T cells to induce colitis. MT-1303 was orally administered to SCID mice every day from day 7 after CD4$^+$ CD45RB$^{high}$ T cell transfer for 21 days, and anti-mTNF-α mAb was intraperitoneally administered to SCID mice on day 7 and day 21 after cell transfer. (A, B) Change in body weight over time. Body weight on each day is expressed as a percentage of the original weight. Each symbol represents the mean ± S.E.M. of body weight (%) in 18 mice. Statistical differences were determined using Student's $t$-test by comparing with the normal group (#$p < 0.05$, ##$p < 0.01$), using Dunnett's test by comparing with the vehicle-treated group (*$p < 0.05$) or using Student's $t$-test by comparing with the vehicle-treated group (†$p < 0.05$, ††$p < 0.01$). (C) Clinical scores were determined on day 28 after cell transfer. Each column represents the mean ± S.E.M. of clinical scores in 18 mice. Statistical differences were determined using the Wilcoxon test by comparing with the normal group (##$p < 0.01$), using Steel's test by comparing with the vehicle-treated group (**$p < 0.01$) or using the Wilcoxon test by comparing with the vehicle-treated group (††$p < 0.01$). There were no significant differences between the anti-mTNF-α mAb-treated group and MT-1303-treated groups using Steel's test.

to less than half of that in the control group (Fig 4B). To examine cytokine production in CD4$^+$ T cells that had infiltrated the colons, LPLs were stimulated with PMA and ionomycin for 6 h, and the number of IFN-γ-producing CD4$^+$ T (Th1) and IL-17-producing CD4$^+$ T (Th17) cells was subsequently determined by intracellular cytokine staining. While there were no obvious differences in the ratio of Th1 or Th17 cells to lamina propria CD4$^+$ T cells between the control group and MT-1303-treated group (Fig 4C), the number of IFN-γ- and IL-17-producing CD4$^+$ T cells in the lamina propria was significantly lower in the MT-1303-treated group than the control group (Fig 4D and 4E). We also measured cytokine concentrations in the culture supernatants of LPLs stimulated with PMA and ionomycin (Fig 4F and 4G). LPLs from MT-1303-treated mice produced significantly less IFN-γ and IL-17 than those from vehicle-treated mice. These results suggest that the amelioration of colitis by MT-1303 is likely due to a reduction of the infiltration of Th1 and Th17 cells into the colon.

## Discussion

Oral administration of MT-1303 at 0.1 and 0.3 mg/kg was efficacious against chronic colitis induced by adoptive transfer of CD4$^+$CD45RB$^{high}$ T cells, a murine IBD model, and its efficacy

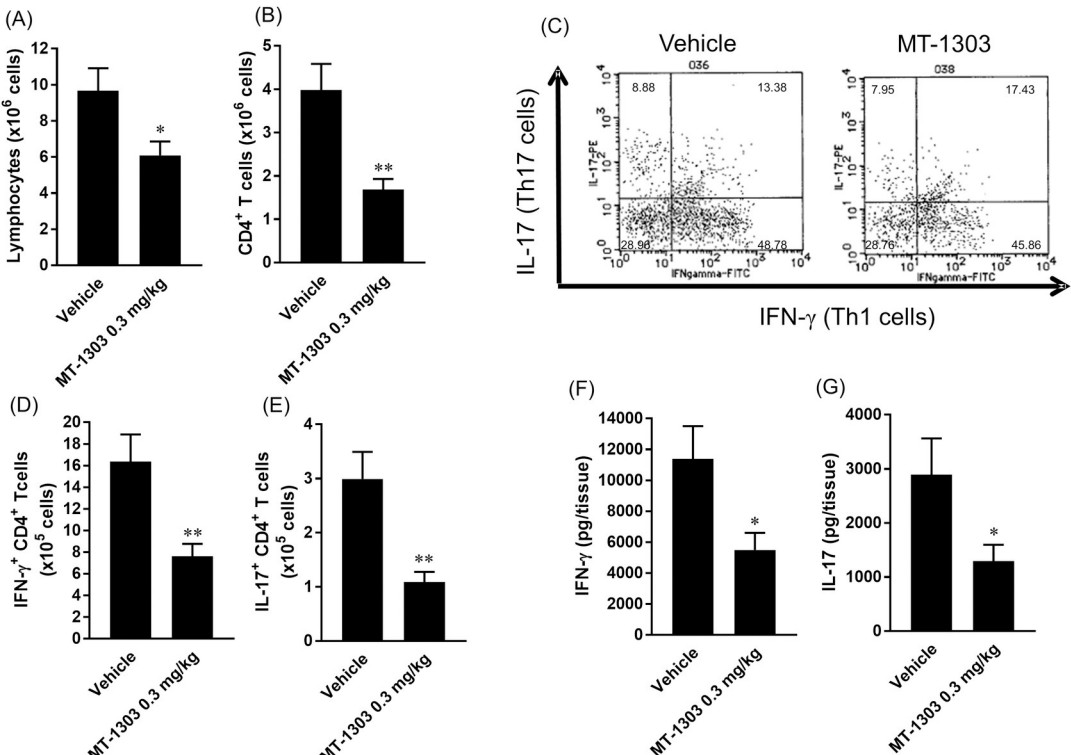

**Fig 4. Effect of MT-1303 on infiltration of Th1 and Th17 cells into the lamina propria of the colon.** MT-1303 or vehicle was orally administered to SCID mice every day from week 1 after CD4[+]CD45RB[high] T cell transfer for 3–4 weeks. (A, B) The number of lymphocytes (A) and CD4[+] T cells (B) in the lamina propria of the colon was measured using flow cytometry the day after the final administration. (C–E) LPLs were stimulated with PMA and ionomycin in the presence of brefeldin A for 6 h before intracellular cytokine staining was performed using anti-IFN-γ and anti-IL-17 mAbs (C). Th1 (D) and Th17 (E) cell counts in the lamina propria of the colon were determined. (F, G) After stimulation with PMA and ionomycin, the amount of IFN-γ (F) and IL-17 (G) in the culture supernatant was determined using a BD[TM] Cytometric Beads Array. Each bar represents the mean ± S.E.M. (n = 11: vehicle-treated group, n = 12: MT-1303-treated group). Statistical differences were determined using Student's $t$-test by comparing with the normal group ($^*p < 0.05$, $^{**}p < 0.01$).

was comparable to that of an anti-mTNF-α mAb. In addition, we confirmed that administration of MT-1303 induced S1P$_1$ internalization in lymphocytes from lymph nodes, prevented the egress of lymphocytes into the periphery and significantly inhibited infiltration of Th1 and Th17 cells into the colon in this colitis model.

We previously reported that the S1P$_1$ agonist activity of MT-1303-P was comparable to that of fingolimod-P [5]. In this study, we compared its S1P$_1$ agonist activity with that of siponimod, ozanimod and etrasimod, second generation S1P$_1$ receptor modulators that are being developed for the treatment of autoimmune diseases. We found that MT-1303-P was the most potent S1P$_1$ receptor agonist, with an EC$_{50}$ value lower than that of these compounds. Fingolimod, a non-selective S1P$_1$ receptor modulator, is known to cause a transient decrease in heart rate that typically occurs within 6 h of the first dose [19]. To reduce such acute negative chronotropic effects, S1P$_3$ receptor-sparing, S1P$_1$ receptor-selective agonists were developed as second generation S1P$_1$ receptor modulators based on preclinical observations in rodents [20,21]. However, because siponimod and ozanimod transiently reduce heart rate in humans, these compounds are initially administered through a dose titration regimen in clinical studies to attenuate acute negative chronotropic effects [22,23]. In contrast, administration of MT-1303 at the anticipated clinical dose does not induce clinically significant bradycardia, nor does it require a dose titration regimen for initiation [4–6].

Both IFN-γ-producing Th1 and IL-17-producing Th17 cells play important roles in IBD pathogenesis and are thought to initiate and promote colitis in humans and mouse models [24,25]. In fact, we confirmed massive infiltration of Th1 and Th17 cells into the colon in this chronic colitis model, and MT-1303 administration decreased the number of infiltrating cells. This suggests that the inhibitory effect of MT-1303 on chronic colitis is mediated by reducing infiltration of colitogenic Th1 and Th17 cells.

CD4+FoxP3+ regulatory T cells (Tregs), in addition to Th1 and Th17 cells, also play an important role in development of colitis [26]. When MT-1303 was administered to C57BL/6 mice daily for 7 days, the number of Tregs (CD4+CD25+FoxP3+) as well as CD4+T cells in peripheral blood was decreased dramatically, although both the number of Tregs and the proportion of Tregs in CD4+T cells were increased in MLNs compared to vehicle-treated control mice (unpublished data). Since MT-1303 treatment influences Tregs trafficking in normal mice, in the future study, it is important to clarify how the administration of MT -1303 affects the number or frequencies of Tregs in colon lamina propria and MLNs in this model.

TNF-α is known to play an important role in mediating gut inflammation in IBD, and anti-TNF-α agents are a well-established treatment option in moderate-to-severe CD and UC. However, around 10–30% of patients are primary non-responders to anti-TNF-α agents and more than 50% of responders develop secondary non-response over time [27,28]. Therefore, new therapeutic agents with a different mechanism of action are urgently required. In this study, we showed that the inhibitory efficacy of MT-1303 on chronic colitis induced by adoptive transfer of CD4+CD45RBhigh T cells was comparable to that of an anti-mTNF-α mAb, indicating that MT-1303 may have therapeutic potential for IBD.

A phase II, multicenter, randomized, double-blind, placebo-controlled trial has been conducted in patients with moderate to severe CD to evaluate the safety and efficacy of MT-1303 (ClinicalTrials.gov NCT02378688), with 78 subjects administered placebo or 0.4 mg of MT-1303 for 14 weeks [29]. Treatment with MT-1303 0.4 mg was generally well tolerated and no new safety concerns related to MT-1303 were reported. However, MT-1303 0.4 mg/day for 12 weeks had no effect on clinical or biochemical disease activity in refractory CD; a high placebo response rate and weaker lymphocyte reduction are thought to have contributed to the negative efficacy result [29].

Overall, this study demonstrated that MT-1303 has comparable efficacy to an anti-mTNF-α mAb against chronic colitis induced by adoptive transfer of CD4+CD45RBhigh T cells. Moreover, it is likely that MT-1303 acts as a functional antagonist of the S1P_1 receptor on lymphocytes and regulates lymphocyte trafficking, thereby inhibiting infiltration of colitogenic Th1 and Th17 cells into the colon. Our nonclinical findings suggest that MT-1303 has potential as a therapeutic agent for the treatment of patients suffering from IBD, including UC and CD.

## Supporting information

**S1 Table. The set of raw data for Table 1.**
(XLSX)

**S2 Table. The set of raw data for Fig 1.**
(XLSX)

**S3 Table. The set of raw data for Fig 2.**
(XLSX)

**S4 Table. The set of raw data for Fig 3.**
(XLSX)

**S5 Table. The set of raw data for Fig 4.**
(XLSX)

## Acknowledgments

The authors acknowledge Kei Sakata, Mitsubishi Tanabe Pharma Corp., for providing preparation of figures.

## Author Contributions

**Conceptualization:** Kunio Sugahara.

**Investigation:** Kyoko Shimano, Yasuhiro Maeda, Hirotoshi Kataoka, Mikako Murase, Sachiko Mochizuki, Hiroyuki Utsumi.

**Supervision:** Koichi Oshita, Kunio Sugahara.

**Visualization:** Kyoko Shimano, Yasuhiro Maeda.

**Writing – original draft:** Kyoko Shimano.

**Writing – review & editing:** Kunio Sugahara.

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
