## [Decision Letter · Decision Letter 0]

30 Aug 2019

PONE-D-19-23451

Amiselimod (MT-1303), a novel sphingosine 1-phosphate receptor-1 functional antagonist, inhibits progress of chronic colitis induced by transfer of CD4+CD45RBhigh T cells

PLOS ONE

Dear Mr. Sugahara,

Thank you for submitting your manuscript to PLOS ONE. After careful consideration, we feel that it has merit but does not fully meet PLOS ONE’s publication criteria as it currently stands. Therefore, we invite you to submit a revised version of the manuscript that addresses the points raised during the review process.

We would appreciate receiving your revised manuscript by Oct 14 2019 11:59PM. To enhance the reproducibility of your results, we recommend that if applicable you deposit your laboratory protocols in protocols.io, where a protocol can be assigned its own identifier (DOI) such that it can be cited independently in the future. For instructions see: http://journals.plos.org/plosone/s/submission-guidelines#loc-laboratory-protocols

We look forward to receiving your revised manuscript.

Kind regards,

Akihiko Yoshimura, PhD

Academic Editor

PLOS ONE

Journal Requirements:

3. At this time, we request that you  please report additional details in your Methods section regarding animal care, as per our editorial guidelines:

(1) Please provide the number of mice used in the study

(2) please include the method of sacrifice

(3) please describe any steps taken to minimize animal suffering and distress, such as by administering analgesics

(4) Please describe the post-operative care received by the animals, including the frequency of monitoring and the criteria used to assess animal health and well-being.

Thank you for your attention to these requests.

4. Please provide additional information about each of the cell lines used in this work, including culture conditions and any quality control testing procedures (authentication, characterisation, and mycoplasma testing). For more information, please see http://journals.plos.org/plosone/s/submission-guidelines#loc-cell-lines.

5. Thank you for stating the following in the Financial Disclosure section:

The authors are employees of Mitsubishi Tanabe Pharma Corporation and received no specific funding for this work.

We note that you received funding from a commercial source: Mitsubishi Tanabe Pharma Corporation

Additional Editor Comments:

Please read reviewers' comments and send revised version.

Reviewers' comments:

Reviewer's Responses to Questions

**Comments to the Author**

1. Is the manuscript technically sound, and do the data support the conclusions?

Reviewer #1: Yes

Reviewer #2: Yes

2. Has the statistical analysis been performed appropriately and rigorously? 

Reviewer #1: Yes

Reviewer #2: Yes

3. Have the authors made all data underlying the findings in their manuscript fully available?

Reviewer #1: Yes

Reviewer #2: Yes

4. Is the manuscript presented in an intelligible fashion and written in standard English?

Reviewer #1: Yes

Reviewer #2: Yes

5. Review Comments to the Author

Reviewer #1: Kyoko Shimano wrote the paper MT1303 S1PR1 antagonist ameliorated experimental colitis model. Previously many papers reported S1PR1 antagonist ameliorated colitis and S1PR1 antagonists are started to use in IBD patient (Clinical trial).

The experiment model was almost enough to support MT1303 suppress the colitis. However I would like to ask some questions below.

First of all, authors need to add the physiological condition. After orally administrated MT1303 to the B6 or Balbc mice, authors need to show the number of Th1/Th17/Treg cells in LP, mesenteric lymph node. Authors also need to add the cell number under the long term administration at least 28days. Show the macro picture of colon and mesenteric lymph node.

Second, in figure3 and 4 authors showed the MT1303 antagonist ameliorated colitis. I would ask the number of Tregs in these experiments, because Tregs are also essential to protect experimental colitis model (Mottet C. JI 2003.).

Finally, I would ask whether MT1303 cure the colitis. To solve the problem, I would ask colitis is ameliorated when MT1303 antagonist is administrated from day 1 to day 14 and observe until day 35 in experimental colitis model.

Reviewer #2: In this manuscript, Shimano et al demonstrated a novel S1P1R antagonist MT-1303 effectively inhibited the development of colitis to prevent infiltrating pathogenic Th1 and Th17 cells into colonic lamina propria. Although it is quite interesting, I have several comments as below.

1. Other than Th1 and Th17 cells, Foxp3+ regulatory T cells (Tregs) are also induced in this colitis model. How Tregs are affected by MT-1303 treatment?

2. A previous publication of the authors (Chiba K et al, Int Immunopharmacol (2003)) showed that the ratio of both Th1 and Th17 in local site (spinal cord) was lower but that in lymph node was higher by FTY720 treatment in EAE model. However, in the colitis model, it seems curious because indeed absolute number of infiltrating T cells was lower, no significant change of the ratio was observed in the intestine. Please explain this discrepancy.

3. It is well known that IL-10 has a regulatory role of immune responses of the intestine. The authors should check whether IL-10 expression in T cells are affected by MT-1303 treatment.

6. PLOS authors have the option to publish the peer review history of their article (what does this mean?). If published, this will include your full peer review and any attached files.

Reviewer #1: Yes: Tomohisa Sujino

Reviewer #2: No

---

## [Author Response · Author response to Decision Letter 0]

4 Oct 2019

We respond to each reviewer's comments on a point-by-point basis in the "Response to Reviewers'".

---

## [Editor Report · Decision Letter 1]

8 Oct 2019

PONE-D-19-23451R1

Amiselimod (MT-1303), a novel sphingosine 1-phosphate receptor-1 functional antagonist, inhibits progress of chronic colitis induced by transfer of CD4+CD45RBhigh T cells

PLOS ONE

Dear Mr. Sugahara,

Thank you for submitting your manuscript to PLOS ONE. After careful consideration, we feel that it has merit but does not fully meet PLOS ONE’s publication criteria as it currently stands. Therefore, we invite you to submit a revised version of the manuscript that addresses the points raised during the review process.

We would appreciate receiving your revised manuscript by Nov 22 2019 11:59PM. To enhance the reproducibility of your results, we recommend that if applicable you deposit your laboratory protocols in protocols.io, where a protocol can be assigned its own identifier (DOI) such that it can be cited independently in the future. For instructions see: http://journals.plos.org/plosone/s/submission-guidelines#loc-laboratory-protocols

We look forward to receiving your revised manuscript.

Kind regards,

Akihiko Yoshimura, PhD

Academic Editor

PLOS ONE

**Journal Requirements:**

**Additional Editor Comments (if provided):**

Both reviewers asked the effect of MT-1303 on Treg cells in the colitis model. However authors did not provide any evidence concerning this point. Unfortunately, no additional experimental data was provided. It is very hard to accept this paper without any additional information. Please perform at least one experiment.

---

## [Author Response · Author response to Decision Letter 1]

19 Nov 2019

According to your comments, we have conducted an experiment to evaluate the effect of MT-1303 on Tregs in normal mice. The results are presented in the rebuttal letter.

---

## [Editor Report · Decision Letter 2]

21 Nov 2019

Amiselimod (MT-1303), a novel sphingosine 1-phosphate receptor-1 functional antagonist, inhibits progress of chronic colitis induced by transfer of CD4+CD45RBhigh T cells

PONE-D-19-23451R2

Dear Dr. Sugahara,

We are pleased to inform you that your manuscript has been judged scientifically suitable for publication and will be formally accepted for publication once it complies with all outstanding technical requirements.

Authors performed experiments showing the effect of MT-1303 on Treg migration. The data are clear and raise a new possibility of the involvement of Tregs in the effect of MT-1303 on anti-inflammatory effects. This paper is now acceptable for publication, but the editor hope that authors further study the effect of this interesting compound on Treg-related long term tolerance in the future.

With kind regards,

Akihiko Yoshimura, PhD

Academic Editor

PLOS ONE

---

## [Editor Report · Acceptance letter]

27 Nov 2019

PONE-D-19-23451R2 

Amiselimod (MT-1303), a novel sphingosine 1-phosphate receptor-1 functional antagonist, inhibits progress of chronic colitis induced by transfer of CD4^+^CD45RB^high^ T cells 

Dear Dr. Sugahara:

I am pleased to inform you that your manuscript has been deemed suitable for publication in PLOS ONE. Congratulations! Your manuscript is now with our production department. 

With kind regards,

on behalf of

Dr. Akihiko Yoshimura 

Academic Editor

PLOS ONE